# Narrative Review and Guide: State of the Art and Emerging Opportunities of Bioprinting in Tissue Regeneration and Medical Instrumentation

**DOI:** 10.3390/bioengineering12010071

**Published:** 2025-01-15

**Authors:** Jaroslava Halper

**Affiliations:** Department of Pathology, College of Veterinary Medicine, The University of Georgia, Athens, GA 30602, USA; jhalper@uga.edu

**Keywords:** tissue engineering, regenerative medicine, 3D printing and bioprinting, biomaterials, bioinks, 4D bioprinting, smart materials, smart stimuli, smart design

## Abstract

Three-dimensional printing was introduced in the 1980s, though bioprinting started developing a few years later. Today, 3D bioprinting is making inroads in medical fields, including the production of biomedical supplies intended for internal use, such as biodegradable staples. Medical bioprinting enables versatility and flexibility on demand and is able to modify and individualize production using several established printing methods. A great selection of biomaterials and bioinks is available, including natural, synthetic, and mixed options; they are biocompatible and non-toxic. Many bioinks are biodegradable and they accommodate cells so upon implantation, they integrate within the new environment. Bioprinting is suitable for printing tissues using living or viable components, such as collagen scaffolding, cartilage components, and cells, and also for printing parts of structures, such as teeth, using artificial man-made materials that will become embedded in vivo. Bioprinting is an integral part of tissue engineering and regenerative medicine. The addition of newly developed smart biomaterials capable of incorporating dynamic changes in shape depending on the nature of stimuli led to the addition of the fourth dimension of time in the form of changing shape to the three static dimensions. Four-dimensional bioprinting is already making significant inroads in tissue engineering and regenerative medicine, including new ways to create dynamic tissues. Its future lies in constructing partial or whole organ generation.

## 1. Introduction

I am not a bioengineer, I am a pathologist and cell biologist interested and pursuing research in tissue repair. I have been fascinated by 3D printing and its possibilities as well as new venues and the promises it provides for the future of medicine. I find the idea that organs can be bioprinted and implanted into human beings very appealing—it would mean that more people can live better lives if they could receive bio- and immunocompatible organs printed on demand and would not have to wait for somebody to die so they can get a chance to live (and take many medications to keep the new organ accepted). The availability of bioprinted organs would solve not only the shortage of donor organs (at least to some degree) but also clarify ethical questions surrounding organ transplantation. When I started working on this review, I realized pretty quickly that this is not going to be a comprehensive review simply because it is such a rapidly evolving field and with so much progress and new technology, a comprehensive review would amount to a book and not to a journal article. I decided to include some of the most exciting and new technologies in the bioprinting field.

Bioengineering, a rapidly developing area of engineering, specializes in creating and/or manufacturing biomaterials that can used in medicine, both human and veterinary, for tissue repair and the reconstruction of structures impaired or destroyed by disease or trauma. Both artificial and natural materials are used in bioengineering, though human-made and non-biological materials have to be biocompatible, able to become part of the newly formed tissue or at least be degradable and non-toxic (and nonimmunogenic if possible). Biomedical engineering concentrates on the development and manufacture of medical devices, including those that can replace organs, such as artificial hearts, pacemakers, and dental and orthopedic prostheses. One can easily envision a situation where it is difficult to distinguish between the two fields. A bioprosthetic Edwards heart valve developed for transcatheter aortic valve replacement is a good example where biological tissue (bovine pericardium) serves as a medium to insert a balloon-expandable cobalt-chromium frame into the human heart via catheter. Though the pericardium is not quite bioengineered, one can envision a scenario where a heart valve would be replaced by a valve consisting not only of bioengineered pericardium but also of other components of the valve as well.

Three-dimensional printing has been widely used and applied to numerous purposes in many industries, including biomedical fields (where it is usually called 3D bioprinting), both for the production of medical devices and for means of tissue regeneration. The evolution of 3D bioprinting took a longer time because of the inherent difficulties connected to special needs, such as the strict sterility of printing equipment, biomaterials, and cells, as well as the maintenance of the viability of cells, biocompatibility, and lack of toxicity just to name a few issues. The more recent appearance of so-called 4D bioprinting where the fourth dimension is time-associated brings new features to bioprinting, enabling changes in the shape of bioprinted items depending on the stimuli encountered in vivo. Organ formation and the production for use in transplantation is one of the ultimate goals of bioprinting as discussed below.

## 2. Three-Dimensional Printing

Three-dimensional (3D) printing, also called additive manufacturing, has been around for more than 40 years, since the early 1980s [1,2]. Products are constructed by the addition of material in layers. An essential part of this process is using computer-aided design (CAD) software modeling, which instructs the 3D printer to create and print the computer-designed object.

Though this process was initially rather slow, it turned out to be suitable for the research and development of new products and the printing of three-dimensional objects. Three-dimensional printing is undergoing a renaissance, with several types of production suitable for a great range of manufacturing of a variety of products by several industries. This technology found application in regenerative medicine, more specifically in orthopedics and dentistry early on [3]. Since then, three-dimensional printing has undergone a lot of progress in terms of methodology and the number of applications in many other fields, including biomedical devices besides those used in tissue regeneration. Though 3D bioprinting implies the creation of biomaterials and tissues, we included a brief section discussing the role of 3D printing in the development and application of medical devices as they are indispensable to bioengineering and as they are (or have the potential to be) used together with bioprinted tissues or be merged with organs and tissues in the body because of their biocompatibility and similarity to tissues such as bone and cartilage. Three-dimensional printing is still developing; for one, it is rather slow and thus not suitable for mass manufacturing, but it is quite versatile, and many modifications are used in many fields and industries. There are certain features common to most if not all types of 3D printing. Three-dimensional printing is easily adaptable in terms of customization, prototyping, and complex geometrics. A variety of materials can be adapted to 3D printing, from metals, thermoplastics, ceramics, fibers, and polymers to biomaterials and the extracellular matrix, the last two with or without cell content. It can be easily customized with prototypes developed in short periods of time, though as pointed out above, the printing itself might be slow. The fast improvements in software and 3D printing facilities that can be accommodated near customers provide definitive advantages when it comes to production itself and to development in techniques and new prototypes and products themselves [1]. For a better understanding of the current applications and future possibilities that may expand the use of bioprinting, I provide a brief summary of the most commonly used techniques and technologies available by 3D printing.

### 2.1. Types of 3D Printing

An ultraviolet (UV) laser used to solidify liquid photopolymer in layers is the basis behind the so-called stereolithography apparatus (SLA). This technique results in solid objects that are highly polished and have great resolution. It is suitable for the creation of detailed prototypes and in the manufacturing of dental and medical equipment using biocompatible resins [4].

Digital light processing (DLP) uses digital light projecting rather than laser to accomplish the same as SLA, but it is much faster, though the resolution is not as good. The speed with additional advantages such as the production of thinner layers and an enlarged number of suitable materials make DLP suitable for jewelry detailing and for dental applications [5].

Fused deposition modeling (FDM) is also known as melt extrusion, where melted multiple layers of hot thermoplastic filaments are extruded through a hot nozzle. Many filamentous materials used widely in bioprinting are processed this way, e.g., polylactic acid, a variety of polymer composites, and ceramic materials, just to name a few examples. Among the main advantages are flexibility and speed of manufacturing. Its biggest applications are in the aerospace and healthcare industries [6].

What makes laser ablation an ablation is the selective removal (i.e., ablation) of a layer of material from a larger block and shaping it into the desired form with a high-power laser beam. This technique turns out to be most useful for the aerospace industry [7].

A high-intensity laser beam is used to polymerize liquid resins (that is the designation of this method as multiphoton polymerization), allowing precise control during several biomedical processes, e.g., tissue engineering, and activities (micro-optics and microfabrication just to name a couple of them) [8].

Inkjet printing has gained importance over time because of its particular importance in 3D bioprinting [9]. In continuous inkjet printing (CIJ), a stream of charged drops is delivered to their final destination by electric or magnetic fields. Drop-on-demand (DOD) inkjet printing is particularly suitable for bioprinting because of its precision in the delivery of individual drops on a substrate. Sterility is easier to maintain with the printing of individual drops rather than streams of drops or filament extrusion. In addition, in DOD, drops are dispensed only when required so there is less wastage of biological fluid. For example, this enables the construction of protein microassays and other low-cost biosensors [10]. Thermal inkjet printing is another version of the inkjet technique suitable for printing a wide range of molecules from DNA to enzymes. The short cycle time between drop formation and the flash heating of the drops preserves the activity of the biomolecules, and thus to low loss of activity of enzymes and other proteins [11]. Two modalities were developed for inkjet bioprinting of actual tissue structures. The first, called cell patterning, uses construction or bioprinting of the substrate on which cells are directly deposited [12]. The second method has a reverse approach: cells are first bioprinted before adhesion or growth-promoting molecules are printed over specific cells. This enables the printing of varying structures over cells growing in a cell culture medium [13]. The review by Saunders and Derby contains many illustrations regarding inkjet printing [9].

Selective laser sintering (SLS) is popular in dentistry and it consists of the laser fusing of tiny particles during the object construction (see more in Section 2.2.3). Selective laser melting (SLM), just like SLS, is an example of a powder bed fusion process where a laser is an energy source for fusing/melting powder materials to transform powder into layers forming the bioprinted part [14]. The process is well illustrated by Sefene (2022) [14].

### 2.2. Three-Dimensional Bioprinting in Biomedical Applications

I do not have to educate readers of this journal about bioengineering and perhaps not even about 3D bioprinting. The development and application of medical devices produced by 3D bioprinting and 3D bioprinting of tissues or their components go together hand-in-hand and are used in many medical fields and specialties.

However, I would like to review the current state of the art of this burgeoning field. Bioprinting can be applied to the printing of tissues using living or viable components, such as collagen scaffolding, cartilage components, and cells, and also to printing parts of structures, such as teeth, using artificial man-made materials that will become embedded in bodies. Many types of components are integrated for numerous applications. Other medical applications may involve the printing of devices, such as prostheses and pacemakers, which are not quite embedded with tissues but are nevertheless integrated with body functions and physiology and may become an integral as well as biodegradable part of relevant tissues and organs sometime in the future.

#### 2.2.1. Plastic and Reconstructive Surgery

This is perhaps the number one field pursuing the development of bioprinting and its use. It has been making advances in the development of branches or techniques of 3D printing to manufacture a multitude of human and animal tissues. A lot of the studies are still in the experimental stage, testing 3D bioprinting of tissues and/or their components, such as skin, nasal and ear cartilage, trachea, and peripheral nerves, on rats and mice [2]. Skin bioprinting is still in the experimental stage, mostly because of the complexity of skin multiple layers, hair follicles, and several types of glands and the variety of cell types participating and embedded in skin composition [2,15,16]. An important aim of many studies is the construction of a decellularized extracellular matrix (ECM), an important scaffold and cell-proliferation stimulating environment [16], leading to faster skin wound healing [17]. Many other challenges have to be solved before skin bioprinting becomes usable in wider medicine, e.g., infections affecting skin burns that can lead to sepsis (and death) and contractures impairing the function of legs and arms. Integrating cellular elements and skin appendages, such as hair follicles and sweat glands, has been shown to be a difficult task as well [2]. So far, most progress has been achieved to some extent with sweat glands [18] and with the advent of 4D printing (see below).

Craniofacial reconstruction has been quite challenging as well due to the anatomical complexities, spatial configurations, and participation of multiple types of tissues and cells in this area. The reconstruction of nasal and auricular components, including cartilage and skin, has been attempted using 3D bioprinting. The reconstruction of nasal cartilage has been studied using a combination of natural and man-made materials [19,20]. Though a lot of progress has been made, most of it is still tested in tissue models and animal trials [2]. Though many biomaterials, including cell-laden lipo-transfer collagen scaffolds, appear to stimulate adipose tissues and the production of growth factors [21], some of them induce rapid degradation and a strong immune response to implants [22].

#### 2.2.2. Orthopedics

Huge progress has been made in repairing and reconstructing previously incorrigible bone defects (e.g., fractures, cancer, and chondromalacia) since the arrival of 3D bioprinting by providing bone grafts and its components, such as bone stents, scaffolds, and ECM [23]. In addition, orthopedics has benefited quite a bit from 3D bioprinting using many biomaterials, such as ceramics, but also calcium phosphate (CaP) and hydroxyapatite as well as titanium and tantalum, and other metals and their alloys that have been used for bone repair and reconstruction with or without bioprinting for some years (see also below under Biomaterials). Overall, 3D bioprinting has proven to be well suited for orthopedics, perhaps because bone regeneration and the use of substitute materials and their production, including bioprinting, has been explored for much longer than for other tissue and perhaps because bone is more amenable to reconstruction by bioprinting metallic biomaterials. Several printing techniques have been utilized with success, such as the extrusion-based technique and SLA. A variety of bioinks and cell types have been useful in promoting osteogenesis [2]. Interestingly, Amler et al. have shown differences in osteogenesis, mineralization, and osteogenic differentiation among mesenchymal progenitor cells derived from different bones [24].

Metallic compounds are eminently suitable for bone repair and reconstruction mostly because of their strength. In many instances, modifications are necessary to ensure that they possess other properties, such as high corrosion resistance, high wear resistance, and low friction [25,26]. These compounds need to be biocompatible (meaning not causing adverse reactions, such as toxicity or rejection), biodegradable so they would not lead to inflammation and other tissue damage, and biomechanical as they need to complement if not equal biomechanical bone parameters [25]. Though many metallic compounds are long-lasting, they do fail, mostly because of wear and corrosion [27].

Perhaps the most commonly used metal is iron because of its ubiquity, biodegradability, and high mechanical properties. To adjust (i.e., decrease) the biomechanical properties and corrosion rate of iron, iron alloys have been developed to prevent stress shielding and to promote osseointegration and bone growth [25]. Intramedullary nails are an integral component of the orthopedic arsenal. It has been shown that iron-based nails are superior to stainless steel nails as the former promotes the differentiation of bone marrow stem cells [28]. Stainless steel has been used a lot in the 3D creation of bone implants, but there are several undesirable effects, such as high-stress shielding, corrosion, and wear resistance, leading to inflammation and allergic reactions due to the leakage of nickel and chromium from stainless steel that can be minimized by its alloys constructed by the addition of titanium, molybdenum, manganese, and/or many others [29]. Such alloys are exquisitely appropriate for manufacturing surgical tools, orthopedic implants, and bone fixation. Three-dimensional bioprinting enables customization for individual patients, and the products are anti-bacterial, easy to clean, and long-lasting [25].

Magnesium and its alloys found their use in manufacturing temporary orthopedic bone tissue implants and devices, including orthopedic screws. They are biocompatible, biodegradable, and lightweight [30].

Ceramics, stable compounds of metals with oxygen or other anions, have been studied for use as orthopedic implants for decades. However, their use was hampered by brittleness and thus sudden unexpected failure [31]. It is only recently that newly developed composites are available as orthopedic implants. New bioactive ceramics, containing hydroxyapatite, are biodegradable and promote bone tissue formation. A similar group of ceramics, the synthetic CaP group containing hydroxyapatite, hyaluronic acid, and tricalcium phosphate, is suitable for use as a bone substitute. It forms a bone-like chemical bond with the living tissue; it is nonimmunogenic, biocompatible, and chemically resembles bone [31]. So-called composite materials found their way into orthopedics as well. They are used to create carbon-enforced polyethererketone implants. Unlike their metal counterparts, they are radiolucent so radiography produces artifact-free images. Because of the lower modulus of elasticity and greater fatigue strength than metal implants, they match bone well and are suitable as implants in the spine and for repair of defects caused by trauma, cancer, and infection [32].

#### 2.2.3. Dentistry

This field, just like orthopedics, is ideally suited for 3D bioprinting, and also the one that has been using 3D printing for a long time. The practical Perceptual Admission Test (PAT), a part of the Dental admission test, is required by many dental schools during entrance exams. PAT evaluates candidates’ spatial visualization skills as dentists need to construct teeth and put together mental images of teeth from X-rays, create and manipulate casts and fillings, and evaluate complicated 2D and 3D objects. All of this can be conducted by 3D printing and 3D imaging to speed up and improve the process of creating new teeth and crowns and their quality among other things. Many dental subspecialties, such as prosthodontics, oral and maxillofacial surgery, and prosthetics have been utilizing and developing new techniques in 3D printing [33,34]. The faithful reproduction of teeth has proven to be of great use and help as surgical guides and in the education of dentists. These are time-saving techniques, bringing care to patients faster and allowing the design of complex implants and teeth with ease in a timely fashion [34].

Several techniques are both popular and very suitable for dentistry and have already been described to some extent above. Stereolithography (SLA) works well for curing polymeric resin where each resin layer is dried and cured before the next layer is applied. Selective laser sintering (SLS) consists basically of laser fusing tiny particles during the object construction. It uses a wide range of materials, from ceramics to metals and polymers. Its biggest plus is the production of high-density materials; however, it requires a large infrastructure [35,36]. A big advantage of FDM is the super-fast hardening of the thermoplastic resin upon extrusion from a nozzle (less than a minute) [34].

Both 3D printing and 3D imaging and modeling are essential in modern dentistry.

#### 2.2.4. Peripheral Nerve Reconstruction

Solving the problem of the limited ability of nerves to regenerate after external injury or degenerative disease has been a major task for neurologists and neurosurgeons alike. Attempts to use nerve autografts and allografts have met with rather limited success [2]. Some development of 3D bioprinted grafts from decellularized ECM and synthetic material to replace injured and/or dead peripheral nerves has been achieved [37]. Another approach for reconstruction combines the use of nerve guidance conduits with stem cells [38]. These innovative venues have been in progress only in laboratories and animal trials so far and human use is still in the future.

#### 2.2.5. Ophthalmology

This is still somewhat mysterious, though it is an absolutely essential medical field for human health and has been developing bioprinting to treat several corneal conditions. Corneal reconstruction with 3D bioprinting has helped with graft development aiming at preserving proper strength, curvature, biocompatibility, and optical clarity, though so far it has not been possible to construct full-thickness cornea [2,39]. One group has been successful in the construction of the corneal stromal layer using collagen-based bioink embedded with primary human corneal stromal keratocytes [40].

#### 2.2.6. Medical Devices

As pointed out above, 3D printing of medical devices has been used with great success for years. The fusion deposition method (FDM) of printing is particularly well suited for this purpose. These devices are not a uniform set. One group includes implants, prosthetics, surgical equipment, and anatomical models among other things that are built layer by layer from digital models by adding materials. Another one is represented by pharmaceutical helpers developed for the controlled release of drugs, including quickly dissolving pills or multi-layered tablets with varying release rates. This is different from products that combine drugs with devices to provide a therapeutic effect. Many such contraptions are widely used by millions of people without realizing that sophisticated 3D printing manufacturing is responsible for the development of drug-eluting stents, pre-filled syringes, inhalers, and transdermal patches. These are examples of pharmaceutical devices used daily by insulin-dependent diabetics, asthma patients, women using certain types of contraception, smokers, or former smokers fighting their nicotine addiction [41].

#### 2.2.7. Organoids

Most cell cultures currently in use are two-dimensional, consisting of cells attached to the bottom of a plastic cell culture plate or flask. They are relatively easy to culture and propagate. Certain, usually hematopoietic, cells tend to grow like small colonies floating in the culture medium. Other cells, hematopoietic and many tumor cells, form colonies in agarose. Such colonies are limited in size and function. They differ in many aspects from so-called organoids, i.e., structures resembling organs. Organoids were developed from pluripotent adult stem cells as 3-dimensional structures possessing some characteristics of their organ of origin, hence the term organoid, which means organ-like. Since organoid cultures were established and characterized, they have been used for drug screening, disease modeling, and efforts to regenerate tissues [42]. Bioprinting turned out to be ideal for the formation of organoids when stem cells are planted on scaffolds and induced to grow as differentiated organoids by the addition of growth factors. Organoids have limitations as they consist of one cell type and they do not have stroma nor are they vascularized [42]. The organization of these cell structures into larger complexes has been challenging; however, utilizing bioprinting has made the formation of larger structures much easier [43]. The following bioprinting techniques are particularly well suited for this task: inkjet-based, laser-assisted, extrusion-based, DLP, and SLA in tandem with a variety of bioinks, e.g., collagen-, alginate-, and gelatin-based, just to give a few examples [42].

### 2.3. Biomaterials

There is a whole slew of biomaterials available for use in 3D printing (additive manufacturing). Some of them are pure biomaterials derived from tissues, extracellular matrix, and cells. They might be used as such, or they might be processed to make them more adaptable for specific use. Polymers are exquisitely suitable for 3D printing as they are versatile and easy to use and they can be derived from biological or man-made materials [1].

#### 2.3.1. Polymers

Natural polymers are abundant as essential components of tissues and organs and indispensable for the proper functioning of living, including, our bodies [44]. Many are used in the manufacturing of paper and textile goods, as food additives, and in nutraceuticals [44]. Their use in cosmetics [45] and drug delivery can be considered biomedical applications [46] but they do not necessarily require 3D bioprinting to be functional. Natural polymers are used together with cells directly in 3D bioprinting. Such polymers are polymeric hydrogels functioning as bioinks to encapsulate cells in the process of 3D printing of tissues, vascular/neural networks, and even entire organs, the Holy Grail of organ transplantation. Their properties of water absorption and retention, biocompatibility, and lack of toxicity contribute to the creation of a stable environment for cells to grow, migrate, proliferate, and/or differentiate and ultimately continue to function as transplanted organs inside bodies. Good examples of such polymers are gelatin, alginate, fibrinogen, and hyaluronic acid. They are water-soluble and all of them are widely available, relatively cheap, and easy to work with [47]. These compounds form bioinks to which cells are added before bioprinting [48,49]. Growth factors are another example of such an addition promoting regeneration and cell proliferation [50,51]. The environment provided by hydrogels enables extracellular matrix development and initial organ formation, including vascular/lymphatic/neural networks upon implantation in vivo [47]. The enormous capacity of hydrogels for water absorption is utilized in manufacturing targeted drug delivery systems where drugs are delivered in small devices that, upon implantation into tissues, are slowly released from the device [52]. Relevant illustrations can be found in Liu et al. [47] and in Silva et al. [44].

Polylactic acid (PLA) is a widely used polymer made from fermenting common sugars that are extracted from corn or sugar cane. PLA is suitable for the manufacture of biomedical products as it is of low toxicity and biodegradable (into lactic acid), which makes it suitable for use as a component of sutures/staples and drug delivery devices, which means that with time, they are degraded and replaced with connective tissue after implantation into the body [53]. PLA (among many other biomaterials) has been used in the healing of recalcitrant and difficult bone defects [54]. Though products such as staples and sutures are routinely manufactured and distributed among medical facilities in large quantities, the ease and versatility of 3D printing would be welcomed by physicians such as plastic surgeons with urgent needs to individualize and modify sutures and staples for their patients.

Flexible polymers, such as thermoplastic elastomers (TPE), thermoplastic polyurethane (TPU), and thermoplastic co-polyesters are widely used in the 3D printing of medical supplies (and other components used in other areas) because of their durability and rubber-like features in addition to the flexibility [1,55].

It is clear from Section 2.2.2 that hydrogels by themselves cannot provide many features necessary for bone formation, including strength, biomechanics, and firmness. The addition of CaP and hydroxyapatite to hydrogel bioinks can compensate partially for the deficiencies pointed out above and supply what is required for the bone to function properly [56]. As pointed out by Tolmacheva et al., the complicated histology and organization of bone requires more than hydrogel bioinks, perhaps the presence of cells, and a mixture of hydroxyapatite and β-tricalcium phosphate. Unsintered or calcium-deficient hydroxyapatite is the best solution for porous scaffolds to promote bone regeneration available at the present time [56]. Inkjet, microextrusion, laser-assisted, and stereolithography bioprinting methods are all suitable for creating scaffolds for use in bone regeneration and repair [57]. This paper by Bogala also contains many illustrations regarding bioprinting methods [57].

#### 2.3.2. Metals

Several metals found their way into other medical fields besides orthopedics as well (see above in Section 2.2.2). For example, iron-based alloys have been used as temporary cardiovascular stents, providing enough strength to keep the blood vessels open. In addition, the biodegradability of these stents encourages the re-endothelialization of blood vessels [25,58]. Several other metals, usually in alloys, have been extensively used for implants; those are stainless steel, magnesium, cobalt, and zinc as described in more detail in Chua et al. [25]. Titanium has been used to construct dental and medical implants, including hip and knee replacements because it is light, biocompatible, not corrosive, and integrates well with bone structures. These qualities are improved when titanium is used in alloys rather than as pure metal [59]. These alloys are also very amenable to 3D printing. Recently, titanium scaffolds were 3D bioprinted to be used for periodontal ligament regeneration [60]. Tantalum, not to be confused with titanium, is used to repair hard tissues, e.g., bones through drug delivery systems, such as dip-coating, surface modification, packaging with hydrogel [61].

### 2.4. Bioinks, Nanocomposite Bioinks, and Other Nanocomposites

Bioink is basically the necessary substance used in bioprinting. It is a combination of biomaterials with cells, though some scientists apply the designation only to biomaterials. These biomaterials have properties suitable for the bioprinting of specific tissues and for accommodation and the high special distribution of specialized cells or at least cells able to differentiate into such cells [49]. Hydrogels are perhaps the most important biomaterials because they facilitate and allow cell support, proliferation, and differentiation. They allow hydration, oxygen, and nutrient permeability and are biodegradable and biocompatible. Biomaterials in the form of nanocomposites have additional advantages over the more “traditional” biomaterials. Nanocomposites added to regular biomaterials even in small quantities enhance surface interactions, viscosity, printability, and biocompatibility among other effects and thus improve cell incorporation into biomaterials and ultimately the printability of these bioinks [49,62]. As mentioned above SLA is used in their production [4]. By necessity, only a brief description of several nanocomposite bioinks follows.

Silicon-based nanobiomaterials are widely used for their ability to strengthen polymeric nanocomposites. Some nanosilicates improve cell viability after printing for up to 120 days [63]. They are well suited to induce osteogenesis, especially when so-called Laponite nanosilicates are added to the cross-linked gelatin methacryloyl-formed nanoengineered ionic-covalent entanglement (NICE) bioinks together with primary bone marrow-derived human mesenchymal stem cells (hMSCs). When printed, UV cross-linked, and incubated in a special medium without growth factors, the NICE bioinks induced encapsulated hMSCs to secrete glycosaminoglycans, proteoglycans, and collagen and this remodeled the bioprinted scaffolds. Amazingly, transcriptome sequencing showed increased expression of numerous genes (*SOX9*, *COL1A2*, *TGFβ2*, *SMAD*, several *BMP*s) upon in vivo implantation of these NICE bioinks [64].

In general, the requirement for successful bone/skeletal application of bioinks is the presence of encapsulated cells, which stimulate cell proliferation and differentiation and promote vascularization (also helped with the addition of vascular endothelial growth factor) [49].

Other uses for silica nanoparticles (SiNPs) have been found in bioimaging and in gene and drug delivery enabled by SiNP regular spherical shapes and large surface area in addition to thermal and mechanical characteristics [49].

Ceramic-based nanomaterials have been shown to be of great use in tissue engineering and regenerative medicine (TERM) as they promote the repair of bone voids and defects, a fairly common but unwelcome event in trauma medicine and orthopedics. Their biocompatibility and osteoconductivity, together with their similarity to natural bone minerals and greater resistance to degradation than hydrogels make them very suitable for use in bone repair [49]. Bioactive glass nanoparticles, belonging to the same class of nanoparticles as ceramic nanomaterial, have shown their usefulness in osteogenesis and bone tissue engineering as well [65].

As mentioned above, CaP nanoparticles belong to a group of related compounds that are very similar to bone and are very useful in bone regeneration. Hydroxyapatite is one of them because of its good bioaffinity and biocompatibility and perhaps mainly because of its similarity to its natural counterpart present in the bone; hence, it is exquisitely suited for use as a component in contrast agent for microcomputed tomography [49]. When incorporated with hydrogels (e.g., alginate and gelatin), the composite bioink was excellent for 3D bioprinting of bone constructs due to its good cell viability (alginate merit), structural stability (provided by gelatin}, and osteoconductivity (presence of hydroxyapatite) [66].

Cellulose-based nanomaterials are derived from cellulose and include nanocellulose, cellulose nanofibrils/nanocellulose fibrils, cellulose nanocrystals/nanocrystalline cellulose, and nanocellulose blends. Nanocellulose is crystalline, with a high specific surface area, surface chemical reactivity, and biocompatibility compound, to name at least some of its characteristics important in bioprinting. It mimics the fibril network of ECM so it serves as reinforcement of structures in bioprinting. These properties make it indispensable as an additive to hydrogels, in the bioprinting of cartilage, tendon, bone, skin, face (plastic surgery), and other tissues and organs [49]. Alginate, a form of cellulose-based material, when combined with nanocellulose fibrils, has found its way into 3D printing of cartilage tissues [67].

## 3. Four-Dimensional Bioprinting

Four-dimensional bioprinting appeared on the scene when it became necessary to incorporate dynamic changes in shape. This requirement led to adding the fourth dimension of time in the form of changing shape to the three existing static dimensions [68]. Five parameters are required to fulfill a successful environment for 4D printing. Those are the additive manufacturing (AM) process, the printing material, the stimuli, the mode of interaction, and the type of modeling [69,70]. Let us very briefly describe the merits of these five parameters. The AM process of printing allows direct manufacturing of print media from the computer without the presence of an intermediate device. A few printing methods are described in the section below. Some of the print media such as smart materials are mentioned throughout the manuscript and also further down as well [71]. The use of a specific smart material determines what stimuli should be used to direct the path of specific transformation of the smart material [72]. The stimuli can be further characterized as physical (light, humidity, magnetic and electrical energy, temperature, and UV light). Chemical stimuli or chemicals can be further divided by their pH and may include both oxidizing substances and reducing media. Biological stimuli involve enzymes and glucose [69,73]. The mode of interaction and its mathematical modeling are the remaining two parameters. The mode of interaction varies with the type of material used and will therefore lead to modification of modeling. This will lead to the determination of the time most favorable to maximize the effect of a particular stimulus on the smart material [74]

Fulfilling these conditions is of particular importance for the medical field during the production of biomimetic tissues, which change shape or function depending on stimuli as they appear with time. Or one might say that in the 4D world, 3D structures become dynamic and change shape or function with time and depending on the nature of stimuli [75]. The discovery and manufacturing of shape-memory materials have been transforming not only biomedicine but also many unrelated industries, such as space, textile, sports, and defense automobile manufacturing [75]. Smart designs for 4D printed structures must be pre-programmed in computer-aided design (CAD) by calculating time-dependent shape changes or deformations of so-called smart materials able to change their geometry in response to external stimuli [76]. Such materials are capable of self-assembly and self-healing, remembering their shapes, and self-reproduction [68,77]. All these properties make these materials ideal to be used in biomedical fields and in particular, in TERM, in drug delivery, and, in the future, in creating living organs suitable for transplantations [52,78].

### 3.1. Printing Methods for Shape-Shifting Scaffolds [78]

Just as with 3D printing, several techniques were developed or modified from 3D versions for 4D printing. Several types of extrusion-based printing found their home in 4D bioprinting: fused deposition modeling (FDM) (see above under 3D printing), direct ink writing (DIW), inkjet printing, SLA, and digital light processing [68] are the most commonly used modalities.

FDM is particularly suitable for printing thermoplastic polymers, not only because of the fast recovery of the shape of compressed scaffolds but because of the induction of shape change in seeded mesenchymal stem cells [79]. This would initiate the proliferation and differentiation of such cells and potentiate if not initiate TERM as well.

Direct ink writing has been found useful for the fabrication of self-healing scaffolds, using urethane diacrylate/polycaprolactone for manufacturing patches for vascular repair [80]. The use of visco-elastic liquid- or paste-like inks is advantageous in a heat-free environment [81]. Some examples of the many materials suitable for this type of printing include nanocomposite polymeric solutions, hydrogels, resins, and several ceramic-based materials. Once printed, the products solidify through a temperature change or using a crosslinking/gelatin mechanism [78]. This method has proven to be very useful for the fabrication of vascular patches using highly stretchable scaffolds that have shape memory and self-healing ability [80].

Similar to direct ink, printing Cryo-3DP transforms liquid printed forms manufactured from hydrogels and other biomaterials into solid stable structures used for the repair of bone defects. The liquid material is printed at −10 °C and compressed to a very small volume, which expands when exposed to near-infrared irradiation [82].

Extrusion-based bioprinting, another form of FDM, is well suited for the printing of bioink-carrying cells. Care must be taken to keep the viscosity not too high so the cells remain viable after extrusion. This is accomplished by the careful increase in applied shear rate and specific printing speed among other parameters [78].

Inkjet-based printing is commonly used to print both biological and non-biological materials (see also above under Section 2.1). It was originally designed for the fabrication of bilayers of gelatin-based materials. When immersed in aqueous media, the scaffolds would self-roll into microtubules mimicking small blood vessels. These structures have the ability to support the implantation and proliferation of endothelial cells [83]. In addition, droplet-based printing was shown to produce self-assembling of cells into structures capable of self-assembly in tissues when stimulated by light [84].

As mentioned above, SLA is used in the manufacturing of bioinks and in dentistry. A UV laser is used to solidify liquid photopolymer in layers where each resin layer is dried and cured before the next layer is applied [4]. As the resin is exposed to air, oxygen may inhibit curing and this may result in incomplete crosslinking and overhanging of the material. In addition, the resin-recoating in the SLA process is rather slow [85].

Digital light processing or DLP resembles SLA but because of its ability to solidify a layer of photopolymer resin during single exposure, it is much faster. This is achieved by the inclusion of a so-called digital micromirror device (DMD) where the micromirrors rotate to direct the light to cure the resin layer in one shot. Both SLA and DLP are nozzle-free systems with fast printing, high printing resolution. and acceptable cell viability [85].

Laser-assisted bioprinting (LAB) is another laser-utilizing printing technique. It is a nozzle-free method using a laser pulse ejecting bioink layer in droplets that contain cells. This is a method with high printing resolution and preservation of cell viability [86].

### 3.2. Smart Biomaterials

The main and most significant contribution of 4D printing lies in the use of stimuli-responsive biomaterials enhanced by stimuli activating the transformation of the printed structure during the post-printing phase. This leads to structural or functional change [78]. Shape memory polymers (SMPs) are perfectly suited for this process (see below). For example, such constructs can be compressed and implanted during a minimally invasive procedure only to inflate to their original and functional size once in situ [82]. For example, nitinol, a shape memory alloy of nickel and titanium, found its way into surgical instruments and implantable medical equipment. Their compressibility before implantation and heat-induced expansion back to their original shape upon implantation of insertion into living tissue have been utilized in many areas, such as orthopedics, vascular stents, and orthodontics [87].

#### 3.2.1. Shape Memory Polymers (SMPs)

These polymers maintain temporary shape and come back to their permanent shape when exposed to external stimuli (heat, magnetic field, stress, and light). That can happen under certain conditions. For example, when the polymer is exposed to a melting temperature, it is deformed and acquires its “permanent” shape. At a temperature below its melting temperature, the material acquires its “temporary” shape [85]. In addition to its shape-morphing capabilities, because some but not all polymers are made from plant materials, these compounds are biocompatible and enable cell viability and growth during and after bioprinting. Synthetic SMPs may actually be cytotoxic, especially during manufacturing when they are exposed to high temperatures and organic solvents [85].

#### 3.2.2. Shape Morphing Hydrogels

In general, hydrogels are water-containing polymers (hence the term hydrogel). They swell or shrink with the amount of water content or change their shape with temperature changes. They are biocompatible and biodegradable, resemble extracellular matrix, and provide a good supportive environment for the proliferation and maintenance of embedded cells [85].

#### 3.2.3. Smart Composite Biomaterials

Sometimes, the addition of other biomaterials is required to overcome certain disadvantageous properties, e.g., limited shape morphing, poor mechanical strength, insufficient printability, and low biocompatibility. Many times, the addition of only small amounts of other biomaterials (e.g., microparticles), especially those called nano-biomaterials like nanofibers (see above), can induce desirable shape changes [88].

### 3.3. Stimuli Involved in Shape Transformation

As already mentioned, those stimuli are necessary to transform shape into its desirable form post-fabrication, meaning after implantation, and usually applied to TERM. As, under the previous subheading there are quite a few items in this category, for the sake of the purpose of this paper, only a few will be described.

**Hydration** is one of the most, if not the most, common forms of inducing shape-change, particularly well-suited for application in our bodies containing a large percentage of water. Hydrogels fit this purpose very well: they are hydrophilic three-dimensional crosslinked polymers. They absorb a lot of water while remaining stable, respond to humidity and temperature, and exhibit memory of previous shapes. Hyaluronic acid and gelatin are two commonly used examples of hydrogels. Many hydrogels are quite biocompatible and non-toxic and are even present in our bodies in substantial quantities, e.g., hyaluronic acid [78].

**Temperature** plays an important role in facilitating shape transformation in **TERM** processes. It is easily applied in TERM conditions. Shape memory polymers respond well to temperature changes. Such polymers can be compressed at a high temperature of 65 °C and expand to their original shape at a lower temperature [78]. Polymers are first compressed to 30% of their original size at a temperature lower than body temperature and expand to their original shape at a body temperature of 37 °C. These characteristics are particularly useful to be used for treating bone defects [89].

**Light** does induce changes in shape, such as uniform cell alignment and tissue maturation, especially with the application of an **NIR** laser [90].

At times, **multiple stimuli** are used to trigger shape changes multiple times but this is beyond the scope of this review [78].

The use of cell-containing bioinks enables the encapsulation of cells in the scaffold, as well as several cell-seeding technologies. Cells can be incorporated into scaffolds post-fabrication and post-printing but before a pre-4D stimulus is applied. Cell incorporation performed post-fabrication can be accomplished with the 3D printing technique and scaffolds with cells implanted this way are useful for drug delivery; they can function as biomolecules or biomedical devices [78]. In the second strategy, cell seeding is performed between printing and pre-4D stimulus. This is useful if the printing would lead to cell damage and would decrease cell viability. The third form of fabrication where cells are incorporated into bioinks pre-printing has the advantage of delivering into the desired place during delivery, but it requires highly cell-friendly bioinks [78].

The possibilities of 4D printing use are many and are expanding. Components for tissue repair and regeneration are similar to those used in 3D printing: scaffolds (or matrix) providing support to cells (usually stem cells or primary cell cultures) and signaling molecules (growth factors and cytokines, mechanical, physical, and electrical stimuli). Cells supported by matrix and stimulated in most cases by multiple stimuli can grow and differentiate, simulating in vivo situations. Smart scaffolds should provide support as close as possible to in vivo matrix support. Drug delivery has a place in 4D bioprinting as well where the use of smart hydrogels facilitates the release of drugs upon the presence of external triggers, such as temperature, pH change, electrical and magnetic fields, light, and drug concentration [52,91]. Obviously, 4D printing of organs suitable for transplantation is still some years away, though the usefulness and applicability of 4D printing in TERM and the creation of organoids, the “babies” of full-fledged organs, represent significant steps forward.

### 3.4. Smart Design

Smart design is necessary to properly direct these smart changes. Because different smart biomaterials undergo changes taking different pathways and exhibiting distinct changes and variations among themselves, bioprinting of different biomaterials, each exhibiting different properties and outcomes, should have distinct printing parameters that would accomplish the desired outcome [85]. Further discussion would be beyond the scope of this review.

### 3.5. Properties of Bioinks Used in 4D Bioprinting

Though I am sure this would be of interest to many readers, the elaboration of the properties of bioinks is again beyond the scope of this review. Suffice to list some desirable properties of these bioinks and a comprehensive review of 4D bioprinting by Lai et al., those properties taken into consideration are biocompatibility, stimuli-responsiveness, biodegradability, mechanical properties, and cells and their viability [85].

### 3.6. Current Applications of 4D Bioprinting

Though this field is similar to 3D bioprinting applications, the development of 4D applications brings the emergence of dynamic newly created constructs and materials for approximating and mimicking the dynamics of living tissues and structures in vivo, a few examples of areas where 4D bioprinting has been shown of use in medicine.

#### 3.6.1. Skin

As pointed out above in 3D bioprinting in biomedical applications, skin bioprinting has been very challenging and not very successful. Four-dimensional bioprinting offers several promising solutions. One consists of fabricated dynamic skin grafts. Another effort produced 4D bioprinted skin capable of changing shape (smart shape skin) adjusting to changes in geometry of healing skin wounds. These new methodologies are utilized both in in vitro and in vivo (in situ) environments [92]. These new models are characterized by cooperation and interactions between skin cells and ECM, an important part of physiological processes in all organs.

#### 3.6.2. Bone

Three-dimensional bioprinting has been useful in enhancing bone regeneration and the repair of bone defects for some years. The use of shape-memory polymers and shape-morphing hydrogels enhanced the maturation of bone scaffolds [93]. Mesenchymal stem cells when cultured in an osteogenic medium can be used in oxidized and methacrylated alginate scaffolds for bone regeneration by undergoing osteogenic differentiation and promoting mineralization [94]. SMPs are used in bone 4D bioprinting in building shape memory scaffolds. Their change in shape also leads to adjustment to irregular bone defects and thus to improved bone regeneration. Bone regeneration is further enhanced by the addition of growth factors and minerals [85]. The 4D-printed bone structures are amenable to change under different stimuli as required [23]. Specialized hydrogels containing 3D cells have been designed to enable flexible function and transformation in response to external stimuli [23,95]. A very good illustration of bone 4D bioprinting is presented by Kang et al. [23].

#### 3.6.3. Cartilage

Progress in creating cartilage has been achieved by 4D bioprinting of cell-laden self-folding scaffolds. Such scaffolds are printed using the DLP methodology. A DLP-created hydrogel bilayer responds to water by folding into a tube-like structure, ready to be used, e.g., as an implant of the trachea (so far tried in a rabbit) [96].

#### 3.6.4. Vasculature

Blood vessels have been constructed by 3D bioprinting for a while. Four-dimensional bioprinting improves precision and flexibility with better control of (small) diameters and complexity of vasculature, especially in the cases of small vessels [85].

#### 3.6.5. Medical Devices

Four-dimensional bioprinting resulted in adding many favorable dynamic properties to devices such as stents, occluders, microneedles, smart 3D-cell engineered microenvironments, drug delivery systems, wound closures, and implantable medical devices, some of which are already mentioned in this paper above. Besides using smart design, shape memory biomaterials, and smart stimuli, the contribution of machine learning and AI will transform the field of 4D bioprinting even more [97].

## 4. Current State and Future Directions of Bioprinting

As described in this review, 3D bioprinting is undergoing great development and evolution in terms of improving the performance of existing printing methods, adding new printing methods, new biomaterials, and synthetic compounds. It is becoming incorporated into many biomedical areas, and it is already indispensable in the manufacturing of individualized dental and orthopedic prostheses and bioprinted bone for the repair of tooth and bone defects. New materials, including metal alloys, allow more biomedical applications to solve more formidable problems, such as formerly recalcitant tissue (not just bone) defects caused by cancer, infections, and/or injuries. In addition, cultures of stem cells together with their development into distinctly differentiated types of cells was one of the first steps leading to in vitro formation and the cultivation of more complex cell structures, including constructs combining stem cells with growing on scaffolds of collagen and extracellular matrix, i.e., organoids.

### 4.1. Near Future Goals

Three-dimensional bioprinting is propelling further developments of these man-made tissue prototypes composed of stem cells grown on acellular scaffolds into complex multilayered tissues, more and more resembling realistic skin and other structures. The preparation of complex tissues for wound implantation and replacement and/or filling tissue defects is coming of age with 3D bioprinting and even more with 4D bioprinting, which enables the shape adjustment of created tissues in response to stimuli, including time. Four-dimensional bioprinting has a great influence on the production of instrumentation and tools, e.g., staples and drug delivery capsules that expand upon tissue or organ implantation.

What else can we expect from bioprinting in the future? For one, the appearance of the fourth dimension greatly expands the range of use of bioprinting. The creation and use of organoids widen horizons for drug testing on organoids simulating specific tissue or organ conditions and thus enhances the quality of in vitro drug testing and limits the use of animals.

Artificial intelligence (AI) can help us to identify new materials and modify printing processing. New biomaterials, hydrogels, and even metal alloys will allow the modification of tissues that might lead to better regeneration and/or function in vivo.

### 4.2. Long-Term Goals and Prospects

The relatively recent modality of 4D bioprinting brings the aim of growing organs in vitro for transplantation closer to reality. The ability of 4D bioprinting to modify shape and even function stimulation in time and/or external circumstances represents a tremendous step forward in creating and assembling organs for transplantation in humans. Such organs could be bioprinted using bionks and “real tissues and cells”; they could be constructed using only synthetic scaffolds and cells, or we would at least be able to combine bioprinted partial organs and tissues with donor tissues. This could be vastly accelerated by using AI and computational chemistry in the identification of novel synthetic replacements for tissues. This would greatly help many very ill people spending less time on the waiting list for an organ donor (who is usually dead) and would represent one or more goals in precision medicine—a brand new organ ready for the recipient who, because of shorter time on the waiting list and because the organ would be custom-made, would have a better chance for a longer and higher qualitatively better life than such patients have today. Future 4D and even higher dimensional bioprinting would become more adaptable to in vivo conditions and would be able to create organs using the recipient’s cells to relegate graft rejection to the past. That would be a significant improvement in terms of patient recovery and long-term prognosis as the immunosuppressive therapy necessary today for the transplant to be accepted by the new host would not be needed anymore, or at least in lower doses. Bioprinted skin would save the lives of many burn patients. One can only imagine that in the rather more distant future, organs with improved physiological functions can be used to “rejuvenate” elderly persons. We can only hope that we will not create new Frankensteins!

## Data Availability

Not applicable.

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
