# Peer review of "Narrative Review and Guide: State of the Art and Emerging Opportunities of Bioprinting in Tissue Regeneration and Medical Instrumentation"

_bioengineering, 2025, doi:10.3390/bioengineering12010071_

Round 1
Reviewer 1 Report
Comments and Suggestions for Authors
This article titled "Narrative Review and Perspective: The State of Art and Emerging Opportunities of Bioprinting in Tissue Regeneration and Medical Instrumentation” reports the modern trends in the field of 3D and 4D medical bioprinting, in particular in tissue regeneration and medical instrumentation. This area is critically important and relevant in the modern world, so such reviews are needed. Unlike typical reviews on this topic, which is focused on one subject over the others, this review is quite specific. It does not contain any illustrations and is more like a handbook on modern areas of bioprinting. This review rather briefly and sequentially reports about all the areas where bioprinting is used giving the references to reviews or significant works in this area, without their detailed discussion. That is why the list of references mainly consists of various reviews of bioprinting in biomedical applications. Due to the very fast development of bioprinting in recent years and the huge number of works in this area, it is very difficult to see the full picture. Nevertheless, this review gives such an opportunity, so I recommend this work for publication in the Bioengineering journal, but after taking into account my comments and suggestions.
The article requires major revision before the consideration for publication in Bioengineering.
The following issues need to be addressed:
1. Since this review is actually a handbook on modern trends in bioprinting, this should be reflected in its title, so that the readers could immediately come up with ​​the main idea of ​​this review.
2. There are no illustrations at all in the review, which makes it very difficult to understand some details. Therefore, I would suggest the author provide at least one illustration representing the main idea in each large subsection.
3. Section 2.1 on 3D printing methods does not report all the methods used in bioprinting. In fact, only polymerization in a bath and polymer extrusion are discussed. Inkjet 3D printing and its various types and applications are not discussed at all. Also, the SLS/M method, which is used to print polymer medical products and products made of biocompatible metals and alloys, is not mentioned here. Although it is mentioned further in Section 2.2.3 but I think it should be mentioned in Section 2.1 and the relevant references should be given.
4. Section 2.2.2 “Orthopedics” is rather poorly covered, although this is probably the most developed and largest area where 3D printing is used. The author should add references to relevant reviews on printing of metal, ceramic and composite implants.
5. Section 2.3 “Biomaterials” reports about polymers, PLA, titanium and tantalum, but nothing is said about such a biomaterial as hydroxyapatite and phosphates. There is one paragraph about hydroxyapatite and its use as filler for hydrogels in Section 2.4. “Bioinks and Nanocomposite Bioinks”. Although, a large number of studies have currently reported the use of hydroxyapatite and calcium phosphates in various fields of biomedicine, including a large number of articles devoted to various types of 3D printing (FDM, SLA and even SLS) of hydroxyapatite-based materials.
6. Section 4 “Future Directions” should be better structured. It should discuss which directions will be actively developed in the coming years, and which ones will follow thereafter, in the distant future, and what the final goal is.
Author Response
Thank you for your constructive and helpful comments and suggestions.
Revisions are in red.
- Since this review is actually a handbook on modern trends in bioprinting, this should be reflected in its title, so that the readers could immediately come up with ​​the main idea of ​​this review.
Revised title: Narrative Review and Guide: The State of Art….
- There are no illustrations at all in the review, which makes it very difficult to understand some details. Therefore, I would suggest the author provide at least one illustration representing the main idea in each large subsection.
As I do not have any illustrations of mine (my research is in a different area) I am pointing out illustrations in specific references – I hope that this is sufficient, find at lines 137, 142, 347, 372, 646.
- Section 2.1 on 3D printing methods does not report all the methods used in bioprinting. In fact, only polymerization in a bath and polymer extrusion are discussed. Inkjet 3D printing and its various types and applications are not discussed at all. Also, the SLS/M method, which is used to print polymer medical products and products made of biocompatible metals and alloys, is not mentioned here. Although it is mentioned further in Section 2.2.3 but I think it should be mentioned in Section 2.1 and the relevant references should be given. Inkjet and SLS/M techniques were added together with references, lines 120-142.
- Section 2.2.2 “Orthopedics” is rather poorly covered, although this is probably the most developed and largest area where 3D printing is used. The author should add references to relevant reviews on printing of metal, ceramic and composite implants. I agree with the reviewer, this section was expanded as suggested.
- Section 2.3 “Biomaterials” reports about polymers, PLA, titanium and tantalum, but nothing is said about such a biomaterial as hydroxyapatite and phosphates. There is one paragraph about hydroxyapatite and its use as filler for hydrogels in Section 2.4. “Bioinks and Nanocomposite Bioinks”. Although, a large number of studies have currently reported the use of hydroxyapatite and calcium phosphates in various fields of biomedicine, including a large number of articles devoted to various types of 3D printing. (FDM, SLA and even SLS) of hydroxyapatite-based materials were added (in 2.2.2. Orthopedics and lines 373-383).
- Section 4 “Future Directions” should be better structured. It should discuss which directions will be actively developed in the coming years, and which ones will follow thereafter, in the distant future, and what the final goal is. That chapter was re-written following your suggestions.
I added a section on organoids (2.2.7).
Reviewer 2 Report
Comments and Suggestions for Authors
Dear Author,
I generally rate the article submitted for review positively. It presents the characteristics of 3D printing, the methods, and materials used for biomedical purposes in a concise form. The very first sentences of the introduction (lines 27-40) show the Author's personal involvement in the discussed issues, which may elicit a positive reaction from the reader. However, I have a few comments, which I am posting below, and it would be good if appropriate corrections were made.
1. Line 97 says “…we provide a brief summary…”. The plural form is used. If there were other co-authors, they should be listed at the beginning of the article.
2. Lines 68-69 contain the same information as line 56. The Author should cross out one of them.
3. The content of chapter 2.2 is suitable for Chapter 1 Introduction
4. I suggest rewriting chapter 4. Usually, the last chapter is conclusions. In this case, it could be the chapter “Conclusions and development prospects”. I suggest that conclusions and Future Directions be written in e-numeric form to be specific and more readable. The first sentence (lines 554, 555) should be crossed out, it repeats the same information for the third time in the article and does not fit the final conclusions.
Kind Regards
Reviewer
Author Response
Thank you for your constructive comments.
Revisions are in green.
I generally rate the article submitted for review positively. It presents the characteristics of 3D printing, the methods, and materials used for biomedical purposes in a concise form. The very first sentences of the introduction (lines 27-40) show the Author's personal involvement in the discussed issues, which may elicit a positive reaction from the reader. However, I have a few comments, which I am posting below, and it would be good if appropriate corrections were made.
- Line 97: “we” changed to “I:
- Lines 68-69 contain the same information as line 56. The Author should cross out one of them. Crossed out on line 56.
- The content of chapter 2.2 is suitable for Chapter 1 Introduction. I did not move chapter 2.2 because I think that Chapter 2 needs introduction as well (it contains several sections).
- I suggest rewriting chapter 4. Usually, the last chapter is conclusions. In this case, it could be the chapter “Conclusions and development prospects”. I suggest that conclusions and Future Directions be written in e-numeric form to be specific and more readable. The first sentence (lines 554, 555) should be crossed out, it repeats the same information for the third time in the article and does not fit the final conclusions- crossed out. Chapter 4 was revised using your and other reviewer’s comments.
Reviewer 3 Report
Comments and Suggestions for Authors
I read this article with interest. I like the style of presentation of information.
I made some comments which can be found in the attached file.
1. There is some problems with using abbreviations. Probably it worth to introduce the list of abbreviations additionally.
2. Probably some subtitles can be added in the text.
3. I do not understand the sentence in the line 276-277.
4. lines 311-314 - not about bio-inks.
5. Probably one or more illustration could be introduced.
Please find additional comments in attached file.

Author Response
Revisions are in blue.
I read this article with interest. I like the style of presentation of information.
I made some comments which can be found in the attached file. Your comments in the pdf file were addressed in the revised documents in Word.
- There is some problems with using abbreviations. Probably it worth to introduce the list of abbreviations additionally. Abbreviations in alphabetical order were added at the end of the text (before references).
- Probably some subtitles can be added in the text. Several new subsections were added with subtitles
- I do not understand the sentence in the line 276-277 changed.
- lines 311-314 - not about bio-inks, title of 2.4 expanded to Bioinks, Nanocomposite Bioinks, and other Nanocomposites
- Probably one or more illustration could be introduced. As I do not have any illustrations of mine (my research is in a different area) I am pointing out illustrations in specific references (in red at lines 137, 142, 347, 372, 646) – I hope that this is sufficient.
I added a section on organoids (2.2.7).
Round 2
Reviewer 1 Report
Comments and Suggestions for Authors
The manuscript was improved and can be published in its current form.